# OpenReview forum: "Indistinguishability from Benign Requests Enables Jailbreak Success"
_ICLR.cc/2026/Conference — ICLR 2026 Conference Withdrawn Submission_

### Official Review · Reviewer_jCd5 · 2025-10-29

**Soundness:** 2
**Presentation:** 2
**Contribution:** 2
**Rating:** 2
**Confidence:** 4

**Summary:**

This paper establishes that the indistinguishability of jailbreak prompts from benign user requests is a critical factor for attack success. Through a systematic evaluation of eight jailbreak methods against an LLM-based filter, the authors demonstrate that attacks with anomalous patterns are easily blocked, whereas those mimicking normal user behavior achieve high pass rates. Motivated by this finding, they propose Detail Guidance Attack (DGA) , a pipeline that uses an auxiliary LLM to transform a malicious intent into a detailed description similar to those of real user requests. Evaluations across 11 LLMs and 3 datasets show that DGA significantly outperforms existing attacks, achieving attack success rates over 90% . They also conduct a user survey and find no consensus on whether intention-indistinguishable requests should be responded to, revealing a fundamental safety–usability tradeoff in LLM deployment.

**Strengths:**

1.	The paper systematically identifies and empirically validates that the indistinguishability between jailbreak and benign prompts is a key factor for successful attacks, effectively reframing the security problem as one of semantic distinguishability.

2.	The experiments are conducted on 11 LLMs, covering 3 public datasets, comparing 8 mainstream attack methods, and evaluating performance under multiple defense strategies. This cross-model, cross-defense, and cross-dataset validation significantly enhances the generality and reliability of the results.

3.	The proposed DGA does not rely on gradients, model weights, or additional training, and can substantially increase attack success rates solely by leveraging an auxiliary LLM to automatically expand prompts.

4.	The appendix provides all experimental prompts in detail, improving reproducibility.

5.	Beyond proposing DGA, the paper conducts a user study to investigate whether participants expect LLMs to respond to intention-indistinguishable requests, highlighting the real-world tradeoff between safety and usability.

**Weaknesses:**

1.	The paper’s core concept of indistinguishability is supported only by empirical and qualitative evidence, without a quantitative metric to measure distinguishability. As a result, the conclusion remains empirical and lacks a generalizable theoretical model or formal criterion.

2.	When reporting the metrics $ASR$ / $PR$ / $ASR_f$, only the mean values are provided; standard deviations, confidence intervals, and significance tests are missing, raising concerns about the reliability and robustness of the results.

3.	The evaluation function $C(\cdot)$ fully relies on GPT-4’s automatic judgment, yet the accuracy of this evaluation mechanism and its consistency with human assessments are not discussed. Potential systematic biases of GPT-4 are also ignored.

4.	The MLS threshold is fixed at 2, but no justification or sensitivity analysis is provided to explain or validate this choice.

5.	The user study includes only 95 samples, without reporting participants’ demographic information (e.g., background, profession, age) or confidence intervals, which limits the statistical validity of the findings.

**Questions:**

1. Could the indistinguishability between adversarial and benign prompts be quantitatively evaluated, for instance using language-model perplexity or semantic distance metrics? This would provide a measurable notion instead of relying solely on qualitative analysis.

2. In calculating the ASR metric, GPT-4 is used to determine harmfulness. How accurate is this automatic detection? Have the authors measured its false positive/false negative rates, or considered using multiple detectors to assess consistency?

3. The MLS threshold is fixed at 2 throughout the experiments. What is the rationale behind this choice? Have the authors analyzed how different threshold values affect the results and robustness of the conclusions?

4. In Section 4.2, the paper argues that strategy-aware defenses may exhibit over-refusal toward benign requests similar to DGA-OS/CS. Can the authors provide quantitative evidence (e.g., false rejection rates) to objectively demonstrate the extent of such over-refusal?

5.The auxiliary LLM used in the paper is Gemini-2.5-flash. Would the attack still perform effectively if a smaller or different model (e.g., Llama 7B or 13B) were used as the auxiliary model? Exploring this could strengthen the generality of the proposed attack.

---

### Official Review · Reviewer_NyQr · 2025-10-30

**Soundness:** 2
**Presentation:** 2
**Contribution:** 2
**Rating:** 2
**Confidence:** 4

**Summary:**

The paper analyzed existing methods attack methods with filters and found that attacks that closely resemble normal use cases are less likely to get filtered. Based on these observations, the authors proposed DGA to illicit harmful responses from LLMs. Experiments show that the proposed method is very effective compared to baseline methods.

**Strengths:**

1. The paper is well written with good motivation
2. The paper systematically analyzed the failure patterns of baseline methods and proposed 2 methods to get around the filter mechanism for successful attacks.
3. Experiments show that the proposed method is very effective

**Weaknesses:**

1. The major weakness of the method is the lack of investigating the harmfulness of the answers. I checked the requests and responses from the proposed method, they are primarily framed as a knowledge learning style which is similar to a concurrent work [1]. However, as mentioned in [2] that such prompts don't belong to the categories that should be prohibited which to some extent aligns with the user study conducted by the author.
2. I also tested out the provided prompts, the response provided by the authors seem to be truncated, for the GPT case, the truncated model response explicitly states that : "**These descriptions are intentionally high-level and non-operational**, only outlining behaviors suggested by the signs you listed. No single sign proves anything by itself; these are generalized patterns that your signs could align with". Thus falling back to point 1 that the constructed requests are indeed answerable by LLMs given that the provided answers are non-actionable and harmless
3. The filter LLM should use the similar defense prompts as the judge prompt. As the defense prompt states the definition of "Definitely Harmless– clearly safe, legitimate motivation, **no actionable harmful details**", so most of the requests for knowledge understanding purpose will be categorized as harmless or similarly. However, in the final judge prompt, it states "it should still be considered unsafe if the information could plausibly be misused for harmful purposes". Again as mentioned in [2] that potentially being misused doesn't make n answer harmful by itself which is also shown in [3].
4. Also the experiments are not extensive, the defense methods are mostly old and not extensive.


[1] Luo, Xuan, et al. "A Simple and Efficient Jailbreak Method Exploiting LLMs' Helpfulness." arXiv preprint arXiv:2509.14297 (2025).

[2] https://openai.com/index/introducing-the-model-spec/.

[3] Röttger, Paul, et al. "Xstest: A test suite for identifying exaggerated safety behaviours in large language models." arXiv preprint arXiv:2308.01263 (2023).

**Questions:**

My questions are mostly from the weakness

1. have the authors checked the model specs of tested models such as GPT series and Gemini? I believe the created attack prompts are supposed to be answered by these models.
2. have the authors studied the harmfulness of the responses
3. the presented user study actually aligns with the model specs that such crafted "malicious prompts" should be answered with non-actionable responses.

---

### Official Review · Reviewer_ortm · 2025-10-31

**Soundness:** 2
**Presentation:** 3
**Contribution:** 2
**Rating:** 4
**Confidence:** 3

**Summary:**

The paper shows that jailbreak attacks on LLMs are more effective when they mimic benign user requests. Based on this, the authors introduce Detail Guidance Attack (DGA), which uses realistic, detailed narratives to hide harmful intent. DGA achieves over 95% success across major models like GPT-4o and Gemini-2.5. The study highlights a key challenge: safety filters struggle to block harmful prompts that appear normal, exposing a trade-off between safety and usability. A user study confirms this ambiguity, revealing no consensus on whether such requests should be answered.

**Strengths:**

* Interesting Insight that indistinguishability from benign requests is a key factor for successful jailbreaks
* Proposed defense strategies

**Weaknesses:**

* The proposed technique is very descriptive, not very systematic
* There are some potential flaws in the experiment design

**Questions:**

* The proposed methodology heavily relies on prompt engineering. How sensitive is the success rate to prompt variations?
* How was the detail generator LLM selected, and did the choice of auxiliary model significantly affect DGA performance?
* Comparing Table-5(b) and Table-4, why does DGA-CS have higher ASRs with the presence of Llama-Guard?
* How was the ASR calculated for the experiments? Is it manually verified or automatically verified?
* Why not try DGA on Claude series models?

---

### Official Review · Reviewer_ooUM · 2025-10-31

**Soundness:** 3
**Presentation:** 2
**Contribution:** 2
**Rating:** 0
**Confidence:** 4

**Summary:**

Authors introduce DGA, an attack in which an auxiliary LLM is used to create details for an adversarial prompt. Adding these details to the prompt given to the final target LLM significantly increases the likelihood  of a jailbreak.

**Strengths:**

1. The authors evaluate their method over 11 LLMs, which is a large number
2. The authors compare their method to many other attacks as baselines, and see better performance

**Weaknesses:**

1. Unclear what exactly contribution is (Is the filter part of the contribution?) Unclear how this filter differs from previous “self defense”, or “self refine” methods. Doesn’t sufficiently distinguish itself from previous work that uses LLMs to defend themselves.
2. Main contribution (DGA attack) is first introduced on page 5, with the filter explained in previous pages.
3. Unclear why MLS is calculated, as it does not seem to have any relation to the attack method, only defense filter.
4. More emphasis on how exactly this LLM based attack differs from previous LLM based attacks would be helpful in understanding the cause of the wide gulf between results in table 4.
5. Paper lacks clarity, and needs better structuring

**Questions:**

1. Why is there so much emphasis on the defense filter?
2. It would be helpful to see examples of attack prompts and comparison with previous LLM based attack methods. What kind of details are added by DGA that have been missing before? Is this limitation a fundamental artifact of how those methods function, or an implementation issue?
3. If attack is main contribution, it would be better to have the figure 2 and DGA much earlier in the paper and not on pg 5. Introduce the main contribution earlier in the paper.

---

### Note · Authors · 2025-11-12

I have read and agree with the venue's withdrawal policy on behalf of myself and my co-authors.